# Metal-Organic Framework-Based Nanomedicines for the Treatment of Intracellular Bacterial Infections

**DOI:** 10.3390/pharmaceutics15051521

**Published:** 2023-05-17

**Authors:** Xiaoli Qi, Ningfei Shen, Aya Al Othman, Alexandre Mezentsev, Anastasia Permyakova, Zhihao Yu, Mathilde Lepoitevin, Christian Serre, Mikhail Durymanov

**Affiliations:** 1Moscow Institute of Physics and Technology, 141701 Dolgoprudny, Russia; 2Faculty of Chemistry, Lomonosov Moscow State University, 119234 Moscow, Russia; 3Institute of Porous Materials from Paris (IMAP), Ecole Normale Supérieure, ESPCI Paris, CNRS, PSL University, 75006 Paris, France

**Keywords:** metal-organic frameworks, nanomedicines, antibiotics, bacteria, intracellular delivery

## Abstract

Metal-organic frameworks (MOFs) are a highly versatile class of ordered porous materials, which hold great promise for different biomedical applications, including antibacterial therapy. In light of the antibacterial effects, these nanomaterials can be attractive for several reasons. First, MOFs exhibit a high loading capacity for numerous antibacterial drugs, including antibiotics, photosensitizers, and/or photothermal molecules. The inherent micro- or meso-porosity of MOF structures enables their use as nanocarriers for simultaneous encapsulation of multiple drugs resulting in a combined therapeutic effect. In addition to being encapsulated into an MOF’s pores, antibacterial agents can sometimes be directly incorporated into an MOF skeleton as organic linkers. Next, MOFs contain coordinated metal ions in their structure. Incorporation of Fe^2/3+^, Cu^2+^, Zn^2+^, Co^2+^, and Ag^+^ can significantly increase the innate cytotoxicity of these materials for bacteria and cause a synergistic effect. Finally, abundance of functional groups enables modifying the external surface of MOF particles with stealth coating and ligand moieties for improved drug delivery. To date, there are a number of MOF-based nanomedicines available for the treatment of bacterial infections. This review is focused on biomedical consideration of MOF nano-formulations designed for the therapy of intracellular infections such as *Staphylococcus aureus*, *Mycobacterium tuberculosis*, and *Chlamydia trachomatis*. Increasing knowledge about the ability of MOF nanoparticles to accumulate in a pathogen intracellular niche in the host cells provides an excellent opportunity to use MOF-based nanomedicines for the eradication of persistent infections. Here, we discuss advantages and current limitations of MOFs, their clinical significance, and their prospects for the treatment of the mentioned infections.

## 1. Introduction

Antibiotics are an important part of modern medicine. However, the treatment of bacterial infections still faces serious challenges owing to antibiotic resistance, which is a rapidly emerging phenomenon. According to a WHO report in 2019, antimicrobial resistance belongs to the top-10 threats to global health, leading to a reduction in antibiotic effectiveness and posing a growing challenge to global healthcare [1,2]. Antibiotic resistance can be achieved by reducing cellular uptake of antibiotics, enhancing their efflux, modifying their metabolism, decreasing their bioavailability, and/or promoting mutations in target genes that make the target insensitive to antibiotics [3].

Among bacterial infections, some pathogens have developed an ability to avoid the host immune response by persistence inside the infected mammalian cells including professional phagocytic cells. Alongside avoiding immune cells, such behavior of bacterial pathogens also provides them with additional protection from antibacterial therapeutics. Indeed, the hydrophilic nature of the most commonly used antibiotics significantly restricts their ability to cross membrane barriers, to accumulate inside the infected cells, and, therefore, to kill intracellular bacteria [4]. The activity of antibiotics is also affected by other factors including pH and enzymatic inactivation. Therefore, their low antibacterial activity inside the cells might result in low therapeutic effectiveness and the emergence of antibiotic resistance [5].

To increase drug accumulation inside the infected cells, numerous nanomedicine platforms have been developed over the last 20 years [1,5]. Similar to bacterial cells, which enter the mammalian cells via endocytic/phagocytic pathways, nanoparticulate carriers can also be internalized via the same route. It means that the use of nanocarriers as a drug delivery system (DDS) might be advantageous for eradication of intracellular pathogens.

Porous coordination polymers, also known as metal-organic frameworks (MOFs), have attracted considerable interest as a promising DDS for antibacterial agents [6,7,8] (Figure 1). MOFs are formed by the self-assembly of metal ions and organic polydentate ligands leading to a wide range of micro- or meso-porous architectures [6,9]. Compared to other nanocarrier systems, MOFs have unique properties, such as highly adjustable chemical composition, high surface area, and tunable pore sizes. Depending on their designed chemical composition and pore size/shape, some of them have low toxicity and are suitable for encapsulating high loadings of numerous therapeutic molecules (including antibacterial agents) with high efficiency and in most cases associated with a prolonged release profile [10,11]. Moreover, MOFs particles’ external surfaces are abundant in functional groups (e.g., amino- and carboxylic groups) that allows for modifying them with targeting moieties or hydrophilic polymers to endow them with stealth properties [9,12,13]. It should be noted that selected MOFs are biodegradable materials with limited in vitro or in vivo toxicity that makes them appropriate for drug delivery [14,15,16].

Here, we aim to discuss antibacterial effects of MOF-based nanomedicines and to overview current MOF formulations against intracellular pathogens including methicillin-resistant *Staphylococcus aureus* (MRSA), *Mycobacterium tuberculosis* (Mtb), and *Chlamydia trachomatis* (Ctr). In addition, we will discuss the advantages of MOF-based nanomedicines and the prospects of their clinical translation for the treatment of the mentioned infections.

## 2. Antibacterial Effects of MOF-Based Nanomedicines

Antibacterial effects of MOF-based nanomedicines can be achieved via different strategies including either MOF structural units (organic and inorganic building blocks) as active agents, or an MOF porous system for therapeutic cargo delivery.

### 2.1. MOFs as Metal Ion Reservoirs

Antibacterial MOFs can provide a source of metal ions, which can be toxic for intracellular bacteria. Controlled degradation of an MOF structure in body fluids results in the release of metal ions from metal nodes and their interaction with bacteria cells.

Among the metals, antibacterial effects have been reported for MOFs containing Ag^+^, Cu^2+^, Co^2+^, Zn^2+^, Fe^3+^, Ni^2+^, Pb^2+^, and others [7,17,18]. The mechanism of antibacterial action may differ from one metal ion to another. For example, the antibacterial features of Ag^+^ ions are related to their ability in breaking the ion balance, destroying the ion channels, and disrupting the integrity of the cell membrane. Cell internalization of Ag^+^ ions leads to their interaction with protein thiol groups, electron chain disturbance, and fragmentation of DNA [19]. The bactericidal effect of Cu^2+^ is based on facilitating lipid peroxidation [20]. Iron-based MOFs can potentiate reactive oxygen species (ROS)-mediated bacteria cell death in acidic conditions due to the Haber-Weiss reaction [21].

Thus, metal ions from the MOF structure can contribute to the antibacterial effects of MOF-based nanomedicines resulting in synergistic effects with other antibacterial payloads.

### 2.2. MOFs Containing Bioactive Linkers

A bactericidal method that relies on a single model is faced with issues such as the need for high doses, limited efficacy against bacteria, or slow sterilization rates. There is a sub-group of MOFs comprising metal clusters coordinated with organic linkers, which hold antibacterial properties and show potential to develop a multi-bactericidal system. In this case, pharmacological substances are incorporated into the MOF skeleton by a one-step synthesis process, and its antibacterial activity is supposed to be improved compared with bare antibacterial molecules thanks to synergistic effects.

Some antibiotics can be used as bioactive linkers for MOF production. For example, antibacterial effects against *S. aureus* have been reported for potassium- and zinc-based MOFs with azelaic acid as a linker [22,23]. Pipemidic acid, a quinolone antibiotic, was used for the production of manganese, zinc, and calcium-based MOFs [24], although their antibacterial activity has not been shown against intracellular pathogens. Moreover, some natural antibacterial agents can also be designed as bioactive linkers. Guo et al. utilized curcumin as a ligand to construct a Zn-MOF-based antibacterial platform, which shows high effectiveness in promoting wound healing for bacterial infections [25].

It is also notable that photosensitizer molecules can be used as organic linkers in MOF structures. Photodynamic therapy (PDT) based on a photosensitizer has shown promising potential in the diagnosis and treatment of infectious diseases by generating ROS under light irradiation and can provide solutions to overcome antibiotic resistance. PDT causes antibacterial effects via damaging bacterial membrane and DNA. Porphyrin and its derivatives have gained significant popularity in PDT. For example, Photofrin (porfimer sodium), a porphyrin-based photosensitizer, was approved by the FDA in 1995 for the treatment of early-stage lung cancer and esophageal cancer, and Levulan (aminolevulinic acid), was approved by the FDA in 2000 for the treatment of actinic keratosis. MOFs show potential for improving the bioavailability of porphyrin, which is vital to develop the next generation of photosensitizers. After the introduction of functional groups to porphyrin, such as carboxylic group and pyridine, it is possible to coordinate them with metal nodes and then produce MOFs with adjustable crystal structures and pore sizes, e.g., PCN-222 (PCN stands for porous coordination network), PCN-223, MOF-525 [26], and PCN-224 [27]. As an example, a study by Zhang et al. [26] demonstrated the feasibility of porphyrin-based metal-organic frameworks to inhibit MRSA growth upon light exposure. Tian et al. developed PCN-224 composite to exert anti-tuberculosis effects based on PDT and immunotherapy, while reducing the incidence of adverse drug reactions and drug resistance [28].

### 2.3. MOFs as a DDS for Antibacterial Agents

In most cases, an MOF porous network is used as a reservoir for antibacterial agents. The amphiphilic microenvironment and tunable porosity of MOFs enables encapsulation of various agents with different antimicrobial mechanisms. Size matching between the payload and pore architecture of MOF pores is essential for successful noncovalent encapsulation. Agents with smaller molecular sizes can be encapsulated in the pores due to π-π stacking, hydrogen bonding, or electrostatic interactions with the framework, in addition to drug-drug intramolecular interactions. It is also possible to consider opposite-charged drugs that are larger than the pore size of MOFs and that will coat the external surface of the nanoMOF [29]. Another strategy consists of synthesizing around an anti-bacterial biomolecule (e.g., Cas9 enzyme) of a size largely exceeding that of the MOFs’ pores, by carrying out a low temperature synthesis in the presence of the biomolecule [30] the MOF particle would form, thus embedding the enzyme or protein within the MOFs’ crystals.

The most common payload of antibacterial MOFs are different antibiotic molecules. MOFs provide a loading capacity of 2–95 wt% for numerous antibiotics including those which are proposed for intracellular pathogens [17]. Sustained release provides for the prolonged antibacterial activity of encapsulated drugs. Antibacterial gases such as nitric oxide can also be co-encapsulated into MOFs with antibiotics as was shown for Ni-based CPO-27 (MOF-74) or MIP-177(Ti) structures. The authors used CPO-27 for the co-delivery of NO and antibiotic metronidazole that resulted in an inhibition of *S. aureus* growth [31], while Ti-MOF with its exceptional chemical stability was used for the prolonged release of NO in body fluids in a variety of wound-healing applications [32]. As well as antibiotics and antibacterial gases, photosensitizers are also used for encapsulation into MOF nanoparticles [33,34]. It cannot be excluded that antibacterial activity of the released photosensitizer can be potentiated by metal ions, which are also involved in ROS production. Similarly, MOF pores are also used for the encapsulation of photothermal molecules [35]. Photothermal therapy (PTT) utilizes photothermal agents to convert near-infrared (NIR) light energy into heat with minimal thermal damage to healthy tissue. As is the case with PDT, PTT is available only for NIR irradiation accessible tissues such as epithelial ones. PTT also can combine PDT, chemotherapy, immunotherapy, and sonodynamic therapy to achieve enhanced therapeutic effect [36,37]. Finally, MOF can also be encapsulated with non-antibiotic antibacterial agents, such as iodine and curcumin [38,39].

Although noncovalent entrapment enables loading hydrophilic, hydrophobic, or amphiphilic agents [14], this inherently reversible drug loading process may cause premature drug release. To circumvent this limitation, covalent post-synthesis attachment to the functional groups on or near the surface of MOFs provides for release of the bioactive agent associated with MOF decomposition [40,41].

All in all, there are a variety of antibacterial mechanisms using MOFs. As a result, this allows for designing a series of materials with combined antimicrobial effects.

## 3. Tuning MOF Properties for Efficient In Vivo Delivery of Antibacterial Agents

Once intracellular bacterial infections initially colonize mucus membranes, followed by reaching distant tissues, physicochemical properties of MOF-based antimicrobial nanomedicines can be tuned for local or for systemic administration. Independent of the route, the particle size of MOFs should not exceed that of the intracellular pathogens that enable the cellular entry of nanoparticles. For this reason, the methodology of MOF production matters a lot. Nanoscale particles can be produced either by microwave-assisted, sonochemical, mechanochemical, or ambient pressure batch synthesis methods, although other techniques can also be applied depending on the chemical MOF structure [42].

In addition to the size, the surface modification of nanoparticles strongly affects the accumulation of MOF nanoparticles in the infected cells and tissues. For example, there is a group of nano-antibiotics designed to target liver and splenic cells of the mononuclear phagocyte system (MPS) in order to kill pathogens which shelter inside Kupffer cells and other resident macrophages [5]. Mannosylation seems to be a simple way for MOF targeting of macrophages expressing mannose receptors (CD206). It was shown in our recent study that modification of the iron fumarate MIL-88A nanoparticles (MIL stands for Materials of Institute Lavoisier) with mannose significantly increased their uptake by alveolar macrophages [43]. At the same time, MOF nanoparticles designed for delivery of antibiotics to Kupffer cells are easily engulfed even without surface functionalization [44].

On the contrary, antimicrobial nanomedicines for targeting infectious foci outside of the liver and spleen should have a prolonged circulation time in the blood because their accumulation in the diseased tissues relies on an “enhanced permeability and retention” effect, mediated by host proinflammatory molecules [45]. As a result, passive accumulation of nanoparticle-encapsulated antibiotics provides for their superior local concentration and sustained release, which cannot be achieved by a systemic administration of free drug. For prolonged blood circulation, the surface of MOF nanoparticles can be modified with polyethylene glycol (PEG) [46,47,48], PEG-grafted dextran [49], platelet-derived membranes [50], or heparin [51]. As well as decreasing MOF recognition by the macrophage, such modification improves colloidal stability and delays the MOF degradation rate [52]. It should be noted that coating with biomimetic membranes might have some limitations including regulatory issues and the presence of unwanted membrane-associated molecules.

Drug release kinetics is another important parameter of antimicrobial MOFs, which has a strong impact on the therapeutic efficacy of MOF-based nanoformulations. Ideally, the release of antibacterial agents from MOF nanoparticles should occur mainly in the cell compartment which shelters a pathogen. Release of the encapsulated drug occurs due to two main mechanisms, including gradient diffusion of the guest molecules from the pores and/or decomposition of the MOF structure [53]. The release of therapeutic ligand molecules incorporated into the MOF structure as organic linkers and metal ions is exclusively mediated by biodegradation. Therefore, the stability of the MOF structure and the drug release behavior in real conditions should be taken into account in respect to an approximate period between MOF nanoparticle administration and cellular uptake.

To achieve precise release profiles with spatial, temporal, and dosage control, multiple stimuli-responsive MOF-based systems have been recently developed [54]. Zeolitic imidazolates or metal carboxylate MOFs exhibit intrinsic pH responsive behaviors with a faster degradation into acidic media for the ZIFs against a faster degradation of the iron or zirconium carboxylate nanoMOFs in blood pH conditions [55]. MOFs containing photosensitizers as organic linkers demonstrate light-responsive properties with either a photothermal or photodynamic effect [56]. Stimuli-responsive drug release and MOF structure decomposition can also be provided by coating the MOF nanoparticles with polymers that are sensitive to enzymatic degradation [26]. Another example is functionalization of MOF nanoparticles with thermoresponsive poly(*N*-isopropylacrylamide) corona. Modified aminated UiO-66 MOFs exhibited controlled release of encapsulated drugs at 40 °C and burst release at 25 °C [57]. As for MOF-mediated drug delivery of antibacterial agents, such microenvironment factors as overexpression of hydrolytic enzymes, decreased pH, and inflammation-induced oxidative stress can serve as endogenous factors, which can be exploited for triggering drug release. Exogenous stimuli such as light and temperature variation can also be used for ‘on-demand’ MOF-mediated drug delivery to superficial infected tissues. It should be noted that the benefits of stimuli-responsive MOF-based DDSs for the treatment of intracellular infections are not obvious because of a very high risk of premature drug release occurring in extracellular space.

Thus, MOF-based nanomedicine designs are determined by multiple factors including administration route, localization of the infected tissue, antibacterial payload, stability of the selected MOF structure in physiological conditions, etc.

## 4. Development of MOFs for the Treatment of Methicillin-Resistant *Staphylococcus aureus* (MRSA)

### 4.1. MRSA Infection and Intracellular Habilitation

Methicillin-resistant *Staphylococcus aureus* (MRSA) is one of the most common drug-resistant pathogens, which was first identified as a clinical isolate from hospitalized patients in the 1960s. However, it has spread rapidly throughout the population since the 1990s [58]. MRSA can spread between humans and animals through direct/indirect contacts such as wounds and medical equipment, and attack a wide variety of organs and tissues, causing severe infections such as boils, cellulitis, endocarditis, chronic osteomyelitis, pneumonia, and bacteremia, thus contributing greatly to global health issues [59,60,61,62]. Once MRSA has developed multiple mechanisms of drug resistance, such as target-site resistance mutations, increased activity of efflux pumps, enzymatic drug modifications, or altered cell wall glycosylation, it can then evade immune recognition, and the treatment of the MRSA infection is extremely challenging [63,64]. For a long time, MRSA has been considered as an extracellular biofilm-forming parasite [61], but in the last decades multiple studies have reported its ability to shelter inside phagocytic and non-phagocytic cells (Figure 2A) [65,66]. MRSA’s ability to occupy an intracellular niche significantly increases pathogen survival under therapy with multiple antibiotics, including methicillin, cephalosporins, macrolides, aminoglycosides, and others [65,67,68]. To date, the most commonly used treatment strategy involves intravenous vancomycin administration. However, even after this treatment, MRSA infections can relapse, resulting in dissemination to distant tissues and the formation of sepsis. This effect was observed in both patients [69] and in experimental animal models [70,71]. It was found to involve intracellular reservoirs of MRSA forms, for instance, in Kupffer cells. In most cases, liver macrophages kill the bacteria, but a minority of the Staphylococci overcome the macrophage’s antimicrobial defenses via several mechanisms [71]. These mechanisms involve overexpression of superoxide dismutase and catalase that directly eliminate ROS [72]. In addition, MRSA produces antioxidants, which scavenge ROS, reactive nitrogen species (RNS), and inhibitors of ROS-producing enzymes of the host cell. Then, the Staphylococci that survived can escape the phagosome and replicate in the cell cytosol, followed by the host cell lysis [73]. In compliance with in vivo data [70,71], intracellular MRSA exhibited increased resistance to vancomycin [74]. Thus, complete eradication of intracellular MRSA remains challenging and requires the development of new therapeutic approaches.

### 4.2. MOF-Based Nanomedicines for MRSA Treatment

MRSA is a widely used experimental in vitro and in vivo infection model for the testing of antibacterial nanomedicines. To date, numerous nanoformulations, including MOF-based ones, have been developed to combat MRSA (Table 1).

Some of the MOF-based nanomedicines or MOF-containing nanocomposites are developed for delivery of antibiotics and toxic metal cations, which can be a part of the MOF structure or be loaded into MOF pores. For example, Huang and colleagues have developed platelet membrane-encapsulated vancomycin (Van)-loaded Ag-based MOF nanoparticles (PLT@Ag-MOF-Van) [50] for the treatment of MRSA infections. It was found that the synthesized PLT@Ag-MOF-Van demonstrated a high drug loading rate, good biocompatibility, and ability to target MRSA-infected areas in mice upon tail vein administration. Coating of MOF nanoparticles with platelet membrane significantly increased their accumulation in the infected lungs as compared with non-coated counterparts. Antibacterial effect of these nanoparticles was mediated due to the combined effect of the released vancomycin and Ag^+^. In the PLT@Ag-MOF-Vanc-treated group, a 100% survival rate was observed, while all mice treated with saline died within 7 days.

In addition to antibiotics and metal cations, other antibacterial payloads are encapsulated into MOF pores. For instance, another study reports using Pluronic-coated MIL-100(Fe) nanoparticles for delivery of encapsulated 3-azido-d-alanine (D-AzAla) to MRSA-infected tissue upon intravenous administration that resulted in D-AzAl release and integration into the bacterial cell wall. Next, the authors injected ultrasmall PEG-coated nanoparticles containing photosensitizer 2-(1-(5-(4-(1,2,2-tris(4-methoxyphenyl)vinyl)phenyl)thiophen-2-yl)ethylidene)malononitrile with dibenzocyclooctyne (DBCO) group, which bound to D-AzAla in bacterial cell wall via click reaction. Further irradiation resulted in significant (75% efficacy) photodynamic inhibition of MRSA and visualization of the infected tissue area [76]. Thus, this two-step approach combines precise in vivo imaging and antibacterial photodynamic therapy.

To provide a precise control of drug release, MOFs and related composites with stimuli-responsive properties have been developed. A study by Song et al. [77] describes construction of zeolitic imidazolate framework (ZIF-8) MOFs modified with a light responsive pH-jump reagent 2-nitrobenzaldehyde (o-NBA), which acts as “gatekeeper” and mediates degradation of MOF structure upon UV irradiation, as an exogenous stimulus, resulting in release of rifampicin (RFP) from the mesopores of ZIF-8. It has been shown that the combination of released Zn^2+^ and RFP from RFP&o-NBA@ZIF-8 could limit MRSA biofilm growth in vitro. In vivo efficacy of these nanoparticles was evaluated in a murine model of an MRSA-infected wound upon local treatment with different MOF formulations with and without UV irradiation. A significant inhibition of MRSA along with accelerated wound healing was observed in the UV and RFP&o-NBA@ZIF-8-treated group in comparison with other groups.

Among endogenous stimuli, decreased pH in the infectious microenvironment, induced by bacterial growth, is widely exploited for designing MOF-based DDS. A pH-responsive MOF-coated mesoporous silica nanoparticle-based co-delivery system (MSN-Sul@carMOF) for carbenicillin (β-lactam antibiotic) and sulbactam (β-lactamase inhibitor) was proposed by Duan et al. [78]. It was found that these nanoparticles can break down biofilms and eliminate MRSA. In addition to antibacterial function, carbenicillin can coordinate Fe^3+^ that prevents sulbactam escape from the pores of MSN. These composite nanocarriers turned out to be non-toxic and capable of simultaneous release of antibiotic and β-lactamase inhibitor in the MRSA-infected tissue upon intravenous injection that resulted in better elimination of antibiotic resistant bacteria in the MRSA-infected skin mouse model and enhanced survival rate in the mouse model of systemic infection induced by MRSA. In the latter case, the amount of MRSA bacteria in the tissues of mice receiving MSN-Sul@carMOF treatment was roughly 13 times lower than that of those treated with MSN-Sul or MSN@carMOF alone. ZIF-8 MOF has been used as a pH-sensitive DDS for delivery of photothermal agent indocyanine green (ICG) [35]. Rapid decomposition of ZIF-8-ICG at decreased pH in the MRSA microenvironment allowed the burst release of Zn^2+^, which increased the permeability of bacterial cell membrane via direct interaction. When exposed to NIR laser irradiation, released ICG produced hyperthermia, which resulted in efficient photothermal bacterial ablation in an MRSA-induced murine subcutaneous abscess model. Chen and colleagues [34] constructed another ZIF-polyacrylic acid-based composite nanomedicine for delivery of photosensitizer methylene blue (MB). MB-loaded ZIF-8-PAA nanoparticles were modified with AgNO_3_/dopamine for in situ reduction of AgNO_3_ to silver nanoparticles (AgNPs), followed by a secondary modification with vancomycin/NH_2_-polyethylene glycol (Van/NH_2_-PEG) to promote surface hydrophilicity. This pH-responsive ZIF-8-PAA-MB@AgNPs@Van-PEG nanoformulation was injected into the vitreous cavity for the efficient treatment of MRSA-induced endophthalmitis in a rabbit model due to the combination of pH sensitivity, the PDT impact of light-activated MB, and the long-acting bactericidal activity of released Ag^+^ and Van.

Another typical hallmark of an infectious microenvironment is the overexpression of hydrolytic enzymes, including hyaluronidase. Zhang et al. [26] manufactured hyaluronic acid (HA)-coated porphyrin-based metal-organic frameworks loaded with Ag^+^ ions (Ag-PCN-224-HA) for combined antibacterial photodynamic therapy and bactericidal effect of released Ag^+^. It was expected that hyaluronidase expressing MRSA will destroy HA coating and induce MOF decomposition. In vivo evaluation of the bactericidal efficacy of Ag-PCN-224-HA was performed under light irradiation using a murine wound model infected with MRSA. The wounds topically treated with PCN-224-Ag-HA and light exhibited complete eschar formation without edema or inflammation as compared to other groups.

In recent years, an increasing number of studies have indicated the involvement of iron as a component of iron-based MOFs in the generation of ROS, which can contribute to antibacterial effect. In this regard, Li et al. [21] created self-activated cascade MOF/enzyme hybrid nanoreactors (MIL@GOx-MIL NRs) containing the mesoporous MIL-101(Fe)-NH_2_, composed of iron (III) and 2-aminoterephtalic acid linker, and GOx (glucose oxidase). GOx was non-covalently incorporated into the MIL shells and onto their surfaces. The embedded GOx could convert glucose to gluconic acid in this system, reducing the pH level from 7.4 to roughly 4.0, where MIL@GOxMIL NRs conduct the maximum cascade reaction activity, including iron-mediated Fenton reaction, and produce a much greater amount of hydroxyl radicals (HO•) than in the same reaction at pH 7.4. The antibacterial impact was then determined using a conventional plate counting test, demonstrating that 5 µg per mL of MIL@GOx-MIL NPs efficiently inhibited the growth of MRSA biofilms. At a concentration of 80 µg per mL of MIL@GOx-MIL NPs that was used, no MRSA biofilms developed. As a result, self-activated cascade reaction nanoreactors of this kind potentially hold promise in the fight against drug-resistant bacteria. A study by Lai et al. [79] describes MIL-101(Fe)-based nanoparticles with covalently attached vancomycin and antimicrobial peptide LL-37 on their surface. It was found that MIL-101(Fe) catalyzes •OH generation via Fenton reaction in the presence of external hydrogen peroxide and acidic pH. Such conditions can be created in extracellular space due to acidification of the surrounding microenvironment stimulated by bacterial growth and inflammatory response of innate immune cells, which produce H_2_O_2_. Alternatively, such conditions are typical for phagocytic compartments, where MRSA can persist. Developed LL-37@MIL-101-Van nanoparticles allow for achieving the synergistic effect of ROS generation and the antimicrobial effects of vancomycin and LL-37 peptide that led to a significant inhibition of biofilm growth and facilitated healing of MRSA-infected wounds in a mouse model after intravenous injection.

It should be noted that it is impossible to evaluate the contribution of antibacterial MOFs to eradicate intracellular MRSA, because most of the studies mentioned here are focused on the examination of integral antibacterial effects of MOF-based nanomedicines in MRSA-bearing mice and do not consider their antibacterial effect against intracellular MRSA. However, it has been shown recently by some of us that MIL-100(Fe) nanoMOFs are able to reach intracellular compartments of *Staphylococcus aureus* infected macrophages and to co-localize with the engulfed pathogen (Figure 2A) [75]. These nanoparticles loaded with two antibiotics (amoxicillin and potassium clavulanate) caused a three to five-fold decrease of bacterial load as compared to free antibiotics added at the same concentrations [75]. Thus, this study demonstrates the feasibility of MOF-based nanomedicines to reach the intracellular reservoir of MRSA and to kill the bacteria. Therefore, the efficiency of other reviewed MOF-based nanomedicines against intracellular MRSA cannot be excluded.

**Table 1 pharmaceutics-15-01521-t001:** MOF-based nanomedicines for MRSA treatment.

MOF-Based Nanomedicine	Antibacterial Mechanism (s)	Infected Mammalian Cell/Animal Model	Antibacterial Effect	Ref.
MIL@GOx-MIL NR (MIL, composed of iron (III) and 2-aminoterephtalic acid linker, and encapsulated glucose oxidase)	MOF as a catalyst of ROS production	-	More than 99.99% inhibition of MRSA biofilm growth	[21]
D-AzAla@MIL-100(Fe) + DBCO-TPETM (Pluronic-coated MIL-100(Fe) encapsulated with 3-azido-d-alanine)	MOF as a DDS for PS precursor	Abscess model in MRSA-infected BALB/C nude mice	In vivo: bacteria-killing efficacy more than 75% after intravenous nanoMOF injection	[76]
LL-37@MIL-101-Van (MIL-101(Fe)-based nanoparticles with covalently attached vancomycin and antimicrobial peptide LL-37)	MOF as a catalyst of ROS production and a DDS for antibiotics	MRSA-infected wounds in Kunming mice	In vitro: ~100% inhibition of MRSA biofilm growth; In vivo: facilitated healing of MRSA-infected wounds after intravenous nanoMOF injection	[79]
ZIF-8-ICG (ZIF-8 MOF loaded with indocyanine green)	MOF as a pH-responsive DDS for PTT	MRSA-induced subcutaneous abscess model in Balb/c mice	In vitro: ~100% inhibition of MRSA biofilm growth; In vivo: more than 93% MRSA ablation after local nanoMOF injection	[35]
ZIF-8-PAA-MB@AgNPs@Van-PEG (ZIF-polyacrylic acid-based NPs loaded with Ag NPs and methylene blue followed by a secondary modification with vancomycin/NH_2_-polyethylene glycol)	MOF as a pH-responsive DDS for PS, antibiotic, and Ag NPs	MRSA-induced endophthalmitis in rabbit model	In vivo: significant MRSA inhibition growth after injection of nanoMOFs into the vitreous cavity	[34]
RFP&o-NBA@ZIF-8 (ZIF-8 MOFs modified with a light responsive pH-jump reagent 2-nitrobenzaldehyde and loaded with rifampicin)	MOF as a UV-responsive DDS for antibiotic	MRSA-infected wound in BALBc mice	In vitro: more than 60% bacterial inhibition rate; In vivo: ~100% MRSA inhibition and accelerated wound healing upon local treatment with nanoMOFs with UV irradiation	[77]
Ag-PCN-224-HA (hyaluronic acid-coated porphyrin-based MOFs loaded with Ag ions)	Stimulus-responsive PS-based MOF as a DDS for Ag ions	Wound model infected with MRSA in Kunming mice	In vitro: more than 90% inhibition of MRSA biofilm growth; In vivo: more than 80% MRSA inhibition and eschar formation without edema or inflammation after topical wound treatment with PCN-224-Ag-HA	[26]
MIL-100(Fe) loaded with amoxicillin and potassium clavulanate	MOF as a DDS for antibiotic	*S. aureus* infected macrophages	In vitro: 3-5-fold decrease of bacterial load as compared to free antibiotics	[75]
MSN-Sul@carMOF (pH-responsive MOF-coated mesoporous silica nanoparticles for carbenicillin and sulbactam)	MOF-containing composite as a pH-responsive DDS for antibiotics	MRSA-infected skin mouse model and mouse model of systemic infection induced by MRSA	In vitro: complete inhibition of biofilm formation; In vivo: enhanced inhibition of MRSA growth and 80% higher rate of mice survival	[78]
PLT@Ag-MOF-Van (platelet membrane-encapsulated vancomycin-loaded Ag-based nanoMOFs)	Ag-based nanoMOFs as a DDS for antibiotic	MRSA-induced lung infection in mice	In vitro: significant inhibition of bacteria growth; In vivo: 100% of mice survival after intravenous injection of PLT@Ag-MOF-Vanc	[50]

## 5. Development of MOFs for the Treatment of Tuberculosis

### 5.1. Mycobacterium tuberculosis Infection and Intracellular Persistence

Tuberculosis (TB) is an infectious chronic disease caused by rod-shaped bacillus *Mycobacterium tuberculosis* (Mtb), which is a highly infectious acid-alcohol-resistant bacilli. Initially Mtb infects alveolar macrophages and avoids digestion in macrophage phagosomes (Figure 3) due to inhibition of phagolysosomal fusion and reduction of oxidative stress [80]. As a result, Mtb exploits macrophages as an intracellular reservoir to avoid immune recognition. Infected macrophages enter the interstitial space and induce inflammatory responses accompanied by recruitment and aggregation of macrophages, dendritic cells, neutrophils, natural killer (NK) cells, and T- and B-cells to the site of the Mtb infection that leads to the formation of the granuloma. Macrophages comprise a core of the granuloma structure surrounded by the other cell populations [81,82]. On one hand, formation of granuloma “encapsulates” the infected macrophages and delays the spread of the pathogen, resulting in latent infection. On the other hand, phagosomal Mtb produces a virulence factor, early secretory antigenic target (ESAT-6), which promotes polarization of macrophages to M2 phenotype. Infected M2-polarized macrophages abundantly secrete interleukin 10 (IL10), which significantly contributes to the development of an anti-inflammatory microenvironment [83] and inhibits T-cell and NK cell-mediated eradication of Mtb in granuloma. Therefore, it remains a serious threat to further Mtb expansion.

Although pulmonary infection is the most common kind of TB, this disease may extend to other parts of the body (extra-pulmonary tuberculosis), including the kidneys, bones, joints, circulatory system, central nervous system, and lymphatic system [84]. Even though TB is preventable and curable, it remains the leading infectious cause of death globally, affecting 10 million and killing 1.4 million people annually [85].

Most anti-TB agents have limited intracellular penetration and poor metabolic stability that leads to inefficient drug delivery and often requires extended treatment durations and/or increased doses of antibiotics [86]. As a result, conventional anti-TB chemotherapy, which includes a course of isoniazid, rifampicin, pyrazinamide, and ethambutol, sometimes causes life-threatening adverse effects and may lead to patient non-compliance [87]. Additionally, TB is the primary cause of mortality among HIV patients and is a significant driver of antibiotic resistance. Particularly, the emergence of drug-resistant TB strains, such as multidrug-resistant TB (MDR-TB) and extensively drug-resistant TB (XDR-TB), is a tremendous hurdle for the treatment of this infection because of their exceedingly low cure rates [86,88].

In response to the global tuberculosis pandemic and the emergence of drug-resistant tuberculosis, effective drug delivery methods have been developed in recent years along with anti-TB drugs.

### 5.2. MOF-Based Nanomedicines for TB Treatment

Using nanocarriers in the treatment of TB holds promise to improve therapeutic efficacy, decrease drug resistance, and enhance bioavailability of therapeutic candidates. As a DDS platform, MOF particles exhibit multiple attractive properties, including the ability to encapsulate multiple therapeutic payloads, as well as intracellular drug delivery to the infected macrophages. This section describes several MOF-based nanomedicines developed for TB therapy.

For example, Simon et al. [89] demonstrated the feasibility of anti-TB medication isoniazid (INH) encapsulation into MIL-100(Fe) with a 13% (*w*/*w*) drug loading. INH@MIL-100(Fe) nanoparticles exhibited a sustained release profile for INH (about 60% released drug for 24 h in PBS, pH 7.4) without a burst release. Similarly, Uthappa et al. [90] designed a hybrid combination of natural diatom biosilica microparticles loaded with MIL-100(Fe) nanoMOFs. This bioinspired rod-like composite exhibited two-fold higher loading capacity for INH and more sustained in vitro drug release as compared with INH-loaded MIL-100(Fe) nanoMOFs. All these findings provide evidence that MIL-100(Fe) is a promising drug delivery platform for INH. In another study, the authors have shown that MIL-101-NH_2_(Fe) MOFs also can be used as a DDS for INH [91]. These MOFs exhibit a similar behavior as MIL-100(Fe) INH in terms of drug loading capacity, albeit a more rapid drug release profile is observed [89,91]. In the follow-up study the authors developed an inhalable spray-dried powder blend of INH-loaded MIL-101-NH_2_(Fe) MOFs and poly(lactide-co-glycolide) (PLGA) [92]. Obtained blends exhibited a prolonged INH release profile as compared to INH-loaded MIL-101-NH_2_(Fe) MOFs. It was also found that exposure of INH-loaded MIL-101-NH_2_(Fe)/PLGA microparticles to RAW246.9 macrophages resulted in higher intracellular drug concentrations than treatment with INH-loaded MIL-101-NH_2_(Fe) particles, a finding that might be a result of enhanced uptake or slower particle degradation.

Another inhalable MOF formulation has been produced via a spray drying method [93]. The authors synthesized MOFs-based copper (Cu^2+^) and polyoxyethylene acetate (POA). POA is a prodrug of the first-line anti-TB medication pyrazinamide (PZA). Obtained spherical particles of Cu(POA)_2_ exhibited a mean diameter of 2.6 µm that made them suitable for inhalation.

Unfortunately, none of the mentioned MOF-based formulations have been tested on infected cell models or animals so far. Lack of therapeutic evaluation studies raises several important concerns regarding the feasibility of this approach. For instance, it has not been shown so far whether these nanomedicines can achieve a phagosomal compartment and co-localize with Mtb upon macrophage uptake. Another concern is the ability of MOF-based nanomedicines to accumulate and penetrate Mtb-infected tissues. Granulomas do not contain a capillary network in the central part, which contains the majority of the Mtb-infected alveolar macrophages. Therefore, regardless of which administration route is used, the MOF-based particles should cross at minimum several cell layers and viscous interstitium to reach a central part of the granuloma. As shown in some tumor spheroid models, MOF nanoparticles can penetrate 3D cell cultures to some extent [94,95]. However, examination of MOF-based particle behavior in granuloma in vitro or in vivo models is highly desirable.

All in all, it is likely that the use of some strategies developed for enhanced nanomedicine penetration in tumors [96] could improve drug delivery and the therapeutic efficacy of MOF-based nanoformulations against TB.

## 6. Development of MOFs for the Treatment of Chlamydial Infections

### 6.1. Chlamydia trachomatis Infection and Life Cycle

*Chlamydia trachomatis* (Ctr) is the most ubiquitous sexually transmitted infection and the leading global cause of infectious blindness. Ctr is a Gram-negative obligate intracellular parasite with a distinct biphasic developmental cycle comprising an infectious elementary body (EB) and a proliferating reticulate body (RB). Upon cellular uptake, EBs transform into metabolically active RBs, which modify a membrane of intracellular vesicular compartment, termed “inclusion”, with bacterial proteins that prevent lysosomal fusion [97]. The remodeled membrane subsequently provides for migration of the inclusion towards the microtubule-organizing center (MTOC), which is near the nutrient-rich peri-Golgi region. Proliferating within inclusions, RBs hijack the nutrients from multiple host cell organelles including Golgi mini-stacks, lipid droplets, mitochondria, peroxisomes, lysosomes, endoplasmic reticulum, endosomes, and multivesicular bodies [98]. Once the growing inclusion occupies most of the host cell volume, the RBs turn back into EBs, which exit the host cell to infect new cells (Figure 4A). Ctr initially infects epithelial cells, followed by further spread to fibroblasts, macrophages, and other cell types.

The treatment of Ctr is still challenging because the invasion of this pathogen into the host cells provides protection from the host immune system and antibacterial drugs. Two membrane barriers, including the plasma membrane and the endosomal/phagosomal membrane of the host cell, significantly restrict the access of antibiotics to Ctr [5,99,100,101]. Moreover, RBs can transform into a metabolically inactive persistent form in response to treatment with antibiotics, pro-inflammatory cytokines (interferon γ), or nutrient deprivation. After elimination of the stressful stimuli, Ctr resumes propagation and dissemination. As well as urogenital tract infections, Ctr causes pneumonia and ocular infections (trachoma) [102]. Therefore, the development of novel antibacterial drugs and antimicrobial drug delivery platforms is crucial for circumventing the defense mechanisms of Ctr.

**Figure 4 pharmaceutics-15-01521-f004:**
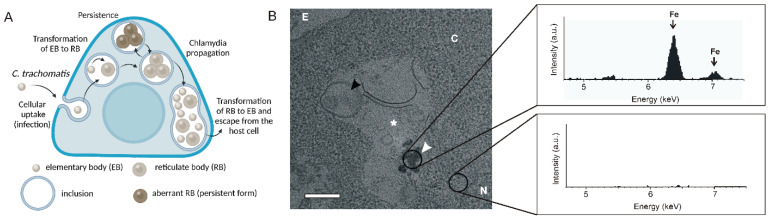
MOF-mediated intracellular delivery of antibacterial agents for treatment of Ctr. (**A**) The infectious cycle of Ctr in cells. Infectious elementary body (EB) internalizes and accumulates in the vesicular compartment, called “inclusion”, where it transforms into a proliferating reticulate body (RB). RBs can transform into a metabolically inactive persistent form. Extensive propagation of the pathogen results in reverse transformation of RB to EB, followed by Ctr escape from the host cell. (**B**) MOF-based nanomedicines can accumulate in chlamydial inclusions. TEM image shows MIL-100(Fe) nanoMOFs co-localized with Ctr in RAW264.7 macrophages. NanoMOFs were determined using Energy Dispersive X-ray Spectroscopy (EDX) spectrum analysis, which showed two major peaks of iron at 6.5 and 7.1 keV from their area, whereas no peaks were detected from the area without nanoparticles. Black and white arrowheads indicate chlamydial EBs and MIL-100(Fe) nanoparticles, respectively. White stars indicate chlamydial inclusions. Abbreviations: N, nucleus; C, cytosol; P, phagosome; E, extracellular space. The scale bar is 500 nm. The figure is adapted with permission from the authors’ pre-print (ref. [102]).

### 6.2. MOF-Based Nanomedicines for the Treatment of Chlamydial Infections

The use of nanoparticles as DDS for the treatment of chlamydial infections is an attractive strategy. Recently, we showed the feasibility of MIL-100(Fe) nanoMOFs to co-localize with Ctr in the infected RAW264.7 macrophages (Figure 4B) [103]. In the same study, a photodynamic strategy was used to eradicate Ctr using photosensitizer-loaded nanoMOFs that could find a potent application for the treatment of mucosal infections. It has been revealed that the nanoMOF that encapsulated the photosensitizer methylene blue (MB) exhibited significant photodynamic inactivation of Ctr in infected cells resulting in a two order of magnitude bacterial burden decrease in comparison with the non-treated infected control and more than a one order of magnitude greater Ctr inhibition in comparison with free MB. Interestingly, it was found that MIL-100(Fe) nanoMOFs have an intrinsic anti-chlamydial effect presumably due to involvement into a Fenton reaction [103]. Thus, our findings suggest the use of iron(III)-based nanoMOFs as a promising drug delivery nanoplatform, which contributes to antibacterial effect, for the treatment of mucus chlamydial infections by local administration. We believe that the developed approach could be valuable for treatment of persistent chlamydial infections because it does not rely on the inhibition of bacteria metabolism as compared with antibiotics.

## 7. Conclusions and Prospects

In recent years, a number of antibacterial nanomedicines have been developed and tested on cell and animal models indicating their improved drug delivery and therapeutic effects in comparison with free drugs [5]. Some of these nanoformulations have been approved for clinical use [104].

To provide an unbiased evaluation of MOF-based nanomedicine prospects for the treatment of intracellular infections, it is necessary to determine their advantages as compared with widely used polymeric and liposomal nanomedicines. It is not possible to utilize one random MOF to deliver many different drugs efficiently compared with other nanomaterials, as the performance of MOFs as DDS can be dramatically influenced by the size and nature of the drug. The unique chemical and physical properties of MOFs can be designed to optimize loading compacity and drug release profiles. Similar to lipid- and polymer-based DDSs, MOF nanoparticles can also internalize and accumulate in close proximity to intracellular pathogens. As for biodistribution behavior upon systemic administration, MOF particles encounter the same concerns as other nanomaterials. However, as opposed to polymers and lipids, MOFs are not simple drug carriers. The ability to incorporate metal ions into the MOF structure allows for imparting them with innate antibacterial toxicity, which does not rely on metabolism interference. As a result, metal ions such as endogenous Fe^2/3+^, Cu^2+^, Mn^2+^, or Zn^2+^ ions might contribute to the antibacterial effect of the drug cargo.

As for the optimal administration route of MOF-based nanomedicines, local administration seems to be a safer and more reliable method for the treatment of intracellular infections. Ctr and Mtb occupy the intracellular niches in mucus tissue and lung alveoli, respectively, which are potentially achievable by local administration. MRSA-induced abscesses or MRSA-infected wounds can also be treated by the local injection of MOF-based nanoparticles, while for the treatment of sepsis only the intravenous route is reasonable. It should be noted that safety requirements for nanomedicines designed for systemic injection are stricter in terms of colloidal stability, size uniformity, and stealth coating. Despite these challenges, the production of MOF-based nanomedicines that meet these requirements is essential for ensuring their clinical effectiveness. However, scaling issues pose a significant obstacle to the cost-effective production of such nanomedicines. The same problem might also be applied to some multicomponent nanoformulations (≥3 components) described here, as a complex multi-stage synthesis procedure always significantly increases the costs and risks of nanoformulation production with each additional step. Therefore, industrial production of such nanomedicines is reasonable only when they exhibit outstanding clinical effectiveness, leading to complete pathogen eradication. In order to achieve this goal, it is necessary to optimize production procedures and explore alternative synthesis methods that are both efficient and cost-effective, while maintaining high standards of safety and efficacy.

Although great progress has been made in the antimicrobial performance of MOFs, the development of treatment platforms based on versatile nanoMOFs is currently in the early phase of investigation. Therefore, much more effort should be made towards clinical translation of MOFs. As mentioned above, MOF nanoparticles can reach the sheltered intracellular bacteria. However, the efficacy of MOF accumulation in bacteria-containing cell compartments remains unclear. To evaluate the sufficiency of bacterial “targeting” and therapeutic effect, multiple studies with diverse in vivo models are desired. Moreover, the development of approaches to enhance the antibacterial selectivity of MOF-based nanomedicines is also necessary. Ideal MOF-based nanoformulations should be nontoxic for mammalian cells, based whenever possible on endogenous metals and/or ligands and/or be easily decomposed and excreted without adverse effects for the host after the killing or inhibition of bacteria. Furthermore, the optimization of current MOF synthesis procedures, and particularly the development of green chemical synthesis, would help to avoid harsh fabrication conditions, as recently reported for the scalable one-pot green ambient pressure preparation of nanoMOFs formulations involving biocompatible MOFs (e.g., MIL-100(Fe), ZIF-8, UiO-66-NH_2_) [105,106], and would eliminate the hazards associated with conventional methods, thereby making MOFs advantageous for industrial production.

Although promising, the clinical use of MOFs to treat infections is still a long-term goal. We are firmly convinced that significant advances in MOF fabrication and cargo loading, as well as tuning their biodegradation behavior and selectivity, will accelerate over the near future for their practical use in the treatment of intracellular infections.

## Figures and Tables

**Figure 1 pharmaceutics-15-01521-f001:**
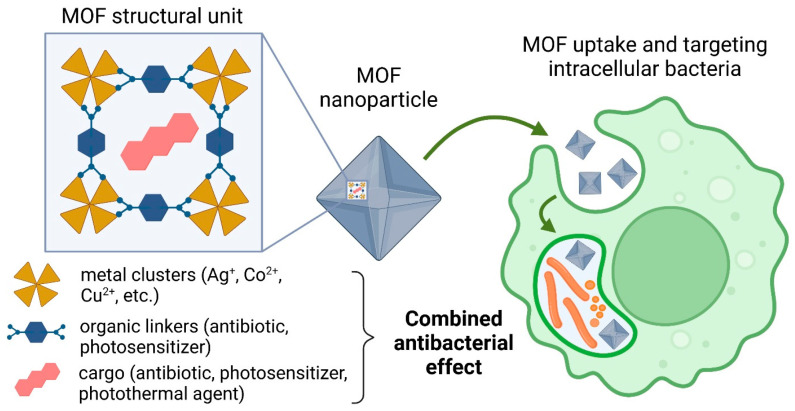
Schematic representation of MOF-based antibacterial nanomedicine for treatment of intracellular infections.

**Figure 2 pharmaceutics-15-01521-f002:**
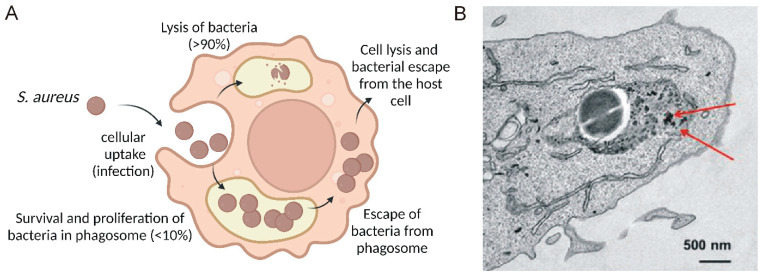
MOF-mediated intracellular delivery of antibacterial agents for the treatment of MRSA. (**A**) The infectious cycle of MRSA in phagocytic cells. Only a minor part of the engulfed bacteria avoids dying in the phagolysosomal compartment due to multiple mechanisms. Further propagation of the pathogen in the vesicular compartment and cytosol eventually leads to lysis of the host cell and bacteria escape. (**B**) MOF-based nanomedicines can accumulate in the vesicular compartment containing the pathogen. TEM image represents MIL-100(Fe) nanoMOFs co-localized with *S. aureus* in J774 macrophages after 1-h incubation (red arrows). The figure adapted with permission from Wiley (ref. [75]).

**Figure 3 pharmaceutics-15-01521-f003:**
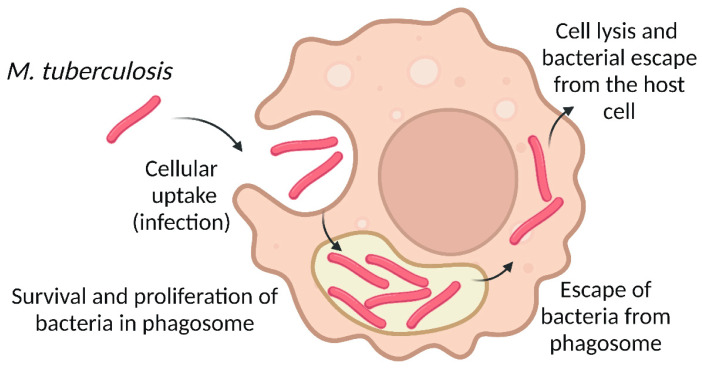
The infectious cycle of Mtb in alveolar macrophages. Mtb avoids digestion in a phagosome due to inhibition of phagolysosomal fusion and reduction of oxidative stress. Further persistence in a vesicular compartment enables bacteria propagation and evasion of immune recognition.

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
