# Peer review of "Metal-Organic Framework-Based Nanomedicines for the Treatment of Intracellular Bacterial Infections"

_pharmaceutics, 2023, doi:10.3390/pharmaceutics15051521_

Round 1

Reviewer 1 Report

The article discusses the potential of metal-organic frameworks (MOFs) as a versatile class of materials for biomedical applications, particularly in antibacterial therapy. It highlights the high loading capacity of MOFs for various antibacterial drugs, their ability to act as nanocarriers for simultaneous encapsulation of multiple drugs, and the presence of coordinated metal ions in their structure, which can increase their innate cytotoxicity for bacteria. The article also covers the numerous functional groups present in MOFs, which enable modification of their external surface for improved drug delivery.

Some comments for the authors:

  1. The review provides examples of MOFs being used for the treatment of intracellular infections such as MRSA, Mtb, and Ctr. Could you also discuss any research or preclinical studies evaluating the efficacy and safety of MOF-based nanomedicines against other intracellular pathogens?
  2. The article mentions that MOFs can be modified with stealth coatings and ligand moieties for improved drug delivery. Could you provide more information on the types of coatings and ligands used and their impact on MOF properties and therapeutic outcomes?
  3. How do MOF-based nanomedicines compare to other existing nanomedicine platforms in terms of biocompatibility, toxicity, and efficacy? A comparison with other nanocarrier systems would provide more context for the reader.
  4. The review briefly mentions the limited in vitro or in vivo toxicity of selected MOFs. Can you expand on the biocompatibility and potential toxicity concerns associated with MOFs, including possible long-term effects and any strategies to mitigate these risks?
  5. The article discusses various strategies for achieving antibacterial effects using MOFs, including their use as metal ion reservoirs. Can you provide more information on the controlled degradation of MOF structures in body fluids and how this process might be tuned for optimal release of metal ions?
  6. In the case of MOFs containing bioactive linkers, what are the factors affecting the stability of these linkers in vivo, and how can the stability be enhanced to ensure the desired therapeutic effect?
  7. What are the key challenges in scaling up the production of MOF-based nanomedicines for clinical use? Can you provide an overview of the manufacturing process and any hurdles that need to be addressed?
  8. The article briefly mentions the possibility of covalent post-synthesis attachment to functional groups on MOFs to prevent premature drug release. Can you provide more details on this approach and any potential drawbacks or limitations?
  9. Are there any known interactions between MOF-based nanomedicines and the host immune system? Can MOFs be designed to modulate immune responses or evade clearance by the immune system?
  10. In the context of the current antibiotic resistance crisis, how do MOF-based nanomedicines address the issue of antibiotic resistance, and what strategies can be employed to minimize the development of resistance against MOF-delivered antibiotics?
  11. The text highlights the importance of surface modification of MOF nanoparticles for targeting specific cells or tissues. Can you provide more examples of targeting ligands or surface modifications that could be employed for the treatment of other intracellular bacterial infections?
  12. You mention that MOF-based nanomedicine designs depend on multiple factors, including administration route and localization of the infected tissue. Could you discuss how these factors influence the choice of MOF structure and surface modifications for efficient drug delivery?
  13. In the section on MRSA treatment, you provide examples of MOF-based nanomedicines encapsulating various antibacterial payloads. Can you discuss the challenges and limitations of encapsulating different types of payloads (antibiotics, metal cations, photosensitizers) within MOF structures?
  14. The article mentions the use of stimuli-responsive MOFs for precise drug release control. Can you provide more details on the design of MOFs with multiple stimuli-responsive properties (e.g., dual or triple responsive) and the potential advantages and challenges of such systems?
  15. The use of light-responsive MOFs for photodynamic therapy is mentioned in the text. Can you discuss the limitations of light penetration in tissue and any strategies that could be employed to overcome these limitations for deep-tissue infections?
  16. In the MRSA treatment section, you mention the use of MOFs for in vivo imaging of infected tissue. Can you provide more examples of MOFs that could be used for simultaneous imaging and therapy (theranostics) in the context of intracellular bacterial infections?
  17. The article describes the use of platelet membrane-coated MOFs for targeted drug delivery to MRSA-infected areas. Can you discuss the potential limitations and challenges of using biological coatings, such as platelet membranes, for MOF-based drug delivery systems?
  18. How do MOF-based nanomedicines compare to other antimicrobial nanoformulations in terms of efficacy, toxicity, and biocompatibility? A comparison with other nanocarrier systems would provide more context for the reader.
  19. The text emphasizes the need for new therapeutic approaches to combat intracellular MRSA. Can you discuss the potential synergistic effects of combining MOF-based nanomedicines with other antimicrobial therapies or strategies to overcome drug resistance in MRSA and other intracellular pathogens?
  20. What are some concerns regarding the feasibility of MOF-based nanomedicines in treating tuberculosis?
  21. In Section 5.1, the description of the Chlamydia trachomatis life cycle is quite detailed. However, it would be beneficial if you could briefly explain the relevance of understanding the life cycle in the context of developing MOFs for the treatment of chlamydial infections.
  22. In Section 6.2, you discuss the use of MIL-100(Fe) nanoMOFs for co-localization with Ctr in infected macrophages. Can you provide more information on the mechanism by which these nanoMOFs are internalized and accumulate in the infected cells? Are there any other MOFs that have shown similar potential for co-localization with intracellular pathogens?
  23. You mentioned the intrinsic anti-chlamydial effect of MIL-100(Fe) nanoMOFs in Section 6.2. Could you expand on the mechanism by which this effect is achieved, and whether other MOFs have displayed similar intrinsic antibacterial properties?
  24. In Section 7, you highlight the advantages of MOFs compared to polymeric and liposomal nanomedicines. However, could you also discuss any potential drawbacks or limitations of using MOFs for the treatment of intracellular infections?
  25. Can you provide more information on the biocompatibility and potential toxicity of MOFs in the context of their use as drug delivery systems, and how these factors may impact their suitability for clinical applications?
  26. Are there any specific strategies that could be employed to enhance the selectivity and targeting of MOF-based nanomedicines towards intracellular pathogens, while minimizing potential off-target effects and toxicity to host cells?
  27. In the discussion of the optimal administration route for MOF-based nanomedicines, you primarily focus on local administration. Could you also discuss the potential challenges and considerations for systemic administration of MOFs in the treatment of intracellular infections?
  28. The article mentions the development of green chemical synthesis for MOFs. Can you elaborate on the potential benefits and limitations of these synthesis methods compared to conventional techniques, and how they might impact the large-scale production and clinical translation of MOFs?
  29. In the conclusion, you mention that the clinical use of MOFs remains a long-term mission. What do you envision as the most critical steps or milestones to be achieved in the near future for the successful translation of MOFs into clinical applications for the treatment of intracellular infections?

None, some minor editing is needed but not a lot

Author Response

Reviewer 1

The article discusses the potential of metal-organic frameworks (MOFs) as a versatile class of materials for biomedical applications, particularly in antibacterial therapy. It highlights the high loading capacity of MOFs for various antibacterial drugs, their ability to act as nanocarriers for simultaneous encapsulation of multiple drugs, and the presence of coordinated metal ions in their structure, which can increase their innate cytotoxicity for bacteria. The article also covers the numerous functional groups present in MOFs, which enable modification of their external surface for improved drug delivery.

Some comments for the authors:

  1. The review provides examples of MOFs being used for the treatment of intracellular infections such as MRSA, Mtb, and Ctr. Could you also discuss any research or preclinical studies evaluating the efficacy and safety of MOF-based nanomedicines against other intracellular pathogens?

Re: To our knowledge, the mentioned pathogens are only those intracellular infections, which have been treated by MOF-based nanomedicines so far.

  1. The article mentions that MOFs can be modified with stealth coatings and ligand moieties for improved drug delivery. Could you provide more information on the types of coatings and ligands used and their impact on MOF properties and therapeutic outcomes?

Re: MOFs can be modified with stealth coatings such as PEG, which can improve their biocompatibility and reduce their clearance by the immune system. The choice of coating depends on factors such as the targetted tissue, administration route, and desired release profile, as described in Chapter 3. All tested MOF nanoformulations are described in detail in chapters 4.2, 5.2, and 6.2.

  1. How do MOF-based nanomedicines compare to other existing nanomedicine platforms in terms of biocompatibility, toxicity, and efficacy? A comparison with other nanocarrier systems would provide more context for the reader.

Re: It is difficult to compare biocompatibility and toxicity of MOF-based nanomedicines with other nanocarrier systems, since these properties strongly depend on the metal and ligand, i.e. MOF structure. It should be noted that we reviewed here low-toxic and biocompatible structures as we mentioned in the Introduction. As for efficacy, we clearly mentioned the main advantage of MOF-based nanomedicines over others in section 7, namely, ability to incorporate metal ions, which impart an innate antibacterial effect to the MOF and potentiate the effect of the drug.

  1. The review briefly mentions the limited in vitro or in vivo toxicity of selected MOFs. Can you expand on the biocompatibility and potential toxicity concerns associated with MOFs, including possible long-term effects and any strategies to mitigate these risks?

Re: Thank you for your suggestion. While we appreciate the importance of discussing the in vitro or in vivo toxicity of MOF-based delivery systems, we believe this discussion is beyond the scope of this review. We think the biocompatibility of MOFs is influenced by many factors such as their size/morphology, metal ions, organic ligands…Briefly, MOFs can induce cellular toxicity and cause tissue damage if they are not properly designed. For example, some MOFs may release toxic metal ions, induce oxidative stress, or trigger inflammatory responses in vivo. However, many studies have shown that MOFs with endogenous metals such as Fe, are preferred due to their low toxicity. In addition, surface modification of MOFs with biocompatible coatings such as PEG might improve their biocompatibility and reduce their toxicity.

  1. The article discusses various strategies for achieving antibacterial effects using MOFs, including their use as metal ion reservoirs. Can you provide more information on the controlled degradation of MOF structures in body fluids and how this process might be tuned for optimal release of metal ions?

Re: Thank you for your suggestion. We think this topic is beyond the scope of this review. The controlled degradation of MOF structures in body fluids has been extensively discussed in reviews on stimuli-responsive systems for drug delivery (https://doi-org.inc.bib.cnrs.fr/10.1002/advs.201801526).

  1. In the case of MOFs containing bioactive linkers, what are the factors affecting the stability of these linkers in vivo, and how can the stability be enhanced to ensure the desired therapeutic effect?

Re: The stability of bioactive linkers in vivo can be influenced by various factors such as enzymatic degradation, pH changes, and redox reactions. Enzymatic degradation can occur due to the presence of enzymes such as proteases or nucleases in the body. pH changes in different environments of the body can also impact the stability of linkers. In addition, redox reactions can occur due to the presence of reactive oxygen species in the body. In addition, surface functionalization of MOFs has been proposed as an ideal strategy to improve their chemical and colloidal stability.

  1. What are the key challenges in scaling up the production of MOF-based nanomedicines for clinical use? Can you provide an overview of the manufacturing process and any hurdles that need to be addressed?

Re: The ambient green synthesis of MOFs-based nanomedicines is very important and interesting to explore (https://doi.org/10.1246/bcsj.20210276). Besides, the large-scale and reproducible synthesis methods, optimization of MOF morphology and size, and standardization of the formulation and quality control processes are also very important to bring MOFs into clinical use.

We emphasized it in the text (lines 534-535): “However, scaling issues pose a significant obstacle to the cost-effective production of such nanomedicines.”

  1. The article briefly mentions the possibility of covalent post-synthesis attachment to functional groups on MOFs to prevent premature drug release. Can you provide more details on this approach and any potential drawbacks or limitations?

Re: Please kindly check the references 40 and 41. And we believe that describing too deeply this part may distract from the key information we are trying to convey in the main body of the review.

  1. Are there any known interactions between MOF-based nanomedicines and the host immune system? Can MOFs be designed to modulate immune responses or evade clearance by the immune system?

Re: Yes. MOFs can be designed to suitable nanoparticles, can serve as carriers for antigens, immune adjuvants, immune agonists, and chemotherapeutic drugs, protecting them from degradation and prolonging their survival in the body, and passively accumulate at tumor sites through the EPR effect, or they can actively target lymph nodes, antigen-presenting cells, or tumor cells through surface modification and size adjustment to reduce the toxic and side effects of exogenous drugs (please, see https://doi-org.inc.bib.cnrs.fr/10.1002/advs.202204932).

  1. In the context of the current antibiotic resistance crisis, how do MOF-based nanomedicines address the issue of antibiotic resistance, and what strategies can be employed to minimize the development of resistance against MOF-delivered antibiotics?

Re: MOFs can act as metal ion reservoirs, releasing metal ions in a controlled manner to achieve antibacterial effects with a low risk of antibiotic resistance generation. Moreover, MOFs can be designed to carry out PDT/PTT, which is highly efficient and considered a promising strategy to replace antibiotics.

  1. The text highlights the importance of surface modification of MOF nanoparticles for targeting specific cells or tissues. Can you provide more examples of targeting ligands or surface modifications that could be employed for the treatment of other intracellular bacterial infections?

Re: We highlighted here all known MOF-based nanomedicines designed for the treatment of intracellular bacterial infections.

  1. You mention that MOF-based nanomedicine designs depend on multiple factors, including administration route and localization of the infected tissue. Could you discuss how these factors influence the choice of MOF structure and surface modifications for efficient drug delivery?

Re: Yes. The choice of MOF structure and surface modifications for efficient drug delivery is highly dependent on the administration route and localization of the infected tissue. For example, for systemic administration, MOF-based nanomedicine with targeting ability and good stability in biological fluids is required. Surface functionalization with biocompatible polymers can improve their circulation time and enhance their targeting specificity to the infected site, also, it can empower MOFs with targeting ability to specific cells or tissues. For example, the use of folic acid-modified MOFs can improve the targeting efficiency of folate receptor-expressing cells. On the other hand, for local administration, MOFs-based nanomedicine with controllable release and high drug-loading capacity may be preferred. These principles of MOF design are discussed in general (section 3) and on the examples (sections 4.2, 5.2, 6.2).

  1. In the section on MRSA treatment, you provide examples of MOF-based nanomedicines encapsulating various antibacterial payloads. Can you discuss the challenges and limitations of encapsulating different types of payloads (antibiotics, metal cations, photosensitizers) within MOF structures?

Re: Encapsulating different types of payloads within MOF structures can present different challenges. For instance, the size, shape, and solubility of the payload molecule can affect its ability to fit within the MOF pore structure, potentially limiting the amount of drug that can be loaded. It is notable to mention that photosensitizers can be sensitive to light and require careful handling encapsulation. The challenges, limitations and prospects are discussed in section 7.

  1. The article mentions the use of stimuli-responsive MOFs for precise drug release control. Can you provide more details on the design of MOFs with multiple stimuli-responsive properties (e.g., dual or triple responsive) and the potential advantages and challenges of such systems?

Re: Please check the answer to question 5.

  1. The use of light-responsive MOFs for photodynamic therapy is mentioned in the text. Can you discuss the limitations of light penetration in tissue and any strategies that could be employed to overcome these limitations for deep-tissue infections?

Re: Yes, one of the major challenges of PDT is the limited penetration depth of light in tissue, which can limit its effectiveness for the treatment of deep-tissue infections. Strategies to overcome this limitation include the use of near-infrared (NIR) light, which has better tissue penetration compared to visible light. This issue is mentioned in section 2.3 (lines 151-152).

  1. In the MRSA treatment section, you mention the use of MOFs for in vivo imaging of infected tissue. Can you provide more examples of MOFs that could be used for simultaneous imaging and therapy (theranostics) in the context of intracellular bacterial infections?

Re: Thanks for your suggestion, but we believe the examples of other types of intracellular bacterial infections lie beyond the scope of this review.

  1. The article describes the use of platelet membrane-coated MOFs for targeted drug delivery to MRSA-infected areas. Can you discuss the potential limitations and challenges of using biological coatings, such as platelet membranes, for MOF-based drug delivery systems?

Re: We added a brief comment on it (lines 192-193).

  1. How do MOF-based nanomedicines compare to other antimicrobial nanoformulations in terms of efficacy, toxicity, and biocompatibility? A comparison with other nanocarrier systems would provide more context for the reader.

Re: Please check the answer to question 3.

  1. The text emphasizes the need for new therapeutic approaches to combat intracellular MRSA. Can you discuss the potential synergistic effects of combining MOF-based nanomedicines with other antimicrobial therapies or strategies to overcome drug resistance in MRSA and other intracellular pathogens?

Re: Please, check citation 28, new therapeutic approaches such as PDT and immunotherapy were combined to exert anti-tuberculosis effects.

  1. What are some concerns regarding the feasibility of MOF-based nanomedicines in treating tuberculosis?

Re: As discussed in the text (lines 437-448), one of the main concerns with using MOF-based nanomedicines for the treatment of tuberculosis is the ability of the MOFs to effectively target and penetrate the granulomas that form in the lungs during infection. Granulomas are complex structures that can limit the delivery of the drugs.

  1. In Section 5.1, the description of the Chlamydia trachomatis life cycle is quite detailed. However, it would be beneficial if you could briefly explain the relevance of understanding the life cycle in the context of developing MOFs for the treatment of chlamydial infections.

Re: As mentioned in this discussion, Ctr can exploit a defensive mechanism via transfer into metabolically inert form, which cannot be treated with antibiotics. In the next section we write that the described MOF/PS-based nanomedicine could be valuable for treatment of persistent chlamydial infections because it does not rely on inhibition of bacteria metabolism as compared with antibiotics.

  1. In Section 6.2, you discuss the use of MIL-100(Fe) nanoMOFs for co-localization with Ctr in infected macrophages. Can you provide more information on the mechanism by which these nanoMOFs are internalized and accumulate in the infected cells? Are there any other MOFs that have shown similar potential for co-localization with intracellular pathogens?

Re: MIL-100(Fe) nanoMOFs are internalized by macrophages via phagocytosis, but it is not yet fully understood how they accumulate in chlamydial inclusions.

  1. You mentioned the intrinsic anti-chlamydial effect of MIL-100(Fe) nanoMOFs in Section 6.2. Could you expand on the mechanism by which this effect is achieved, and whether other MOFs have displayed similar intrinsic antibacterial properties?

Re: This effect is presumably due to the Fenton reaction, as mentioned in lines 500-501.

The reaction can be written as follows:

Fe2+ + H2O2 → Fe3+ + OH• + OH-

In this reaction, ferrous ion reacts with hydrogen peroxide to produce ferric ion, hydroxyl radical, and hydroxide ion. For other iron-based MOFs, it would display similar effects.

  1. In Section 7, you highlight the advantages of MOFs compared to polymeric and liposomal nanomedicines. However, could you also discuss any potential drawbacks or limitations of using MOFs for the treatment of intracellular infections?

Re: We mentioned in the text (lines 534-535) that keeping up the safety requirements upon scaling is the main issue, which is common for all nanomedicines including MOF-based ones.

  1. Can you provide more information on the biocompatibility and potential toxicity of MOFs in the context of their use as drug delivery systems, and how these factors may impact their suitability for clinical applications?

Re: Please check the answer to question 3.

  1. Are there any specific strategies that could be employed to enhance the selectivity and targeting of MOF-based nanomedicines towards intracellular pathogens, while minimizing potential off-target effects and toxicity to host cells?

Re: Actually, this is a key challenge as we mentioned in section 7 (lines 546-550). The only possible way is to maximize cellular uptake of MOF-based nanomedicines by infected cells that could be achieved by surface functionalization with targeting ligands. However, there is no reliable approach that can help to specifically recognize the infected cells.

  1. In the discussion of the optimal administration route for MOF-based nanomedicines, you primarily focus on local administration. Could you also discuss the potential challenges and considerations for systemic administration of MOFs in the treatment of intracellular infections?

Re: Please check the answer to question 12.

  1. The article mentions the development of green chemical synthesis for MOFs. Can you elaborate on the potential benefits and limitations of these synthesis methods compared to conventional techniques, and how they might impact the large-scale production and clinical translation of MOFs?

Re: Please check the answer to question 7.

  1. In the conclusion, you mention that the clinical use of MOFs remains a long-term mission. What do you envision as the most critical steps or milestones to be achieved in the near future for the successful translation of MOFs into clinical applications for the treatment of intracellular infections?

Re: The most critical steps in our opinion are mentioned in the text (lines 543-558).

Reviewer 2 Report

Major Comments

In general, the manuscript is interesting in using metal-organic frameworks (MOFs) to produce nanomedicines to treat intracellular bacterial infections. However, some general comments are provided to the authors in order to improve the manuscript. The authors describe several MOFs applying as nanomedicine against three different types of bacteria. However, to increase their understanding, it will be helpful to provide one or two figures explaining and summarising the different kinds and uses of these MOFs. In the conclusions, the authors briefly comment on the cost of producing MOFs. It will be better to explain this regard in one section and whether these MOFs are now used in some illnesses in humans or not. The last is one of several questions that generate to the readers.

Menor Comments

In addition, the authors should revise some comments by sections of the manuscript to improve it, as follow:

Topic 5.1.

Line 387, the authors should correct the classification of Mycobacterium tuberculosis as Gram-positive bacteria because this microorganism shows another kind of cell wall that differs clearly from those Gram-stained bacteria. Mycobacteria are classified as Acid-alcohol-resistant bacilli. Therefore, it should be noted that Gram stain is unhelpful in identifying Mycobateria.

Line 407, the authors should add a reference at the end.

Topic 5.2.

Line 433, separate “a13”.

Lines 446 to 449, revise the size of letter.

Topic 6.

Line 472, fix “5.1” for 6.1.

Line 505, the “EDX spectrum analysis” is not introduce in the text, at least explain the abbreviantion.

Author Response

Reviewer 2

Major Comments

In general, the manuscript is interesting in using metal-organic frameworks (MOFs) to produce nanomedicines to treat intracellular bacterial infections. However, some general comments are provided to the authors in order to improve the manuscript. The authors describe several MOFs applying as nanomedicine against three different types of bacteria. However, to increase their understanding, it will be helpful to provide one or two figures explaining and summarising the different kinds and uses of these MOFs. In the conclusions, the authors briefly comment on the cost of producing MOFs. It will be better to explain this regard in one section and whether these MOFs are now used in some illnesses in humans or not. The last is one of several questions that generate to the readers.

Re: Thank you for your comments. To increase understanding and combine the comments from other reviewers, we have added one figure to explain the MOF-based antibacterial nanomedicine. Please check the updated Figure 1.

We appreciate your suggestion to provide more detailed information on the cost of producing MOFs and their current usage in human illnesses. While great progress has been made in the antimicrobial performance of MOFs, it is currently in the early phase of research, to the best of our knowledge, there are currently no MOF-based drugs or therapies that have been approved for use in humans. As well as the cost of production, intracellular mechanism understanding, safety and efficacy evaluation are also important. So, we decided not to create a new section to highlight it. Nevertheless, we have made some revisions to ‘Conclusions and Prospects’ part to clarify any potential confusion.

Minor Comments

In addition, the authors should revise some comments by sections of the manuscript to improve it, as follow:

Topic 5.1.

Line 387, the authors should correct the classification of Mycobacterium tuberculosis as Gram-positive bacteria because this microorganism shows another kind of cell wall that differs clearly from those Gram-stained bacteria. Mycobacteria are classified as Acid-alcohol-resistant bacilli. Therefore, it should be noted that Gram stain is unhelpful in identifying Mycobateria.

Line 407, the authors should add a reference at the end.

Topic 5.2.

Line 433, separate “a13”.

Lines 446 to 449, revise the size of letter.

Topic 6.

Line 472, fix “5.1” for 6.1.

Line 505, the “EDX spectrum analysis” is not introduce in the text, at least explain the abbreviation.

Re: All the mentioned points have been revised, thanks for your comments.

Reviewer 3 Report

In this review article, the authors discussed the recent advancements in metal-organic framework-based nanomedicines for treating MRSA, TB, and Ctr infections. The authors also showed MOFs as metal ion reservoirs and drug delivery systems for antibacterial agents. Overall, the review was well written and should be accepted after considering including the following comments.

1. It would be helpful to the readers if a general scheme for making MOFs is described in a dedicated figure.

2. Can authors include a dedicated section on current limitations for MOFs and perspective at the end?

Author Response

Reviewer 3

In this review article, the authors discussed the recent advancements in metal-organic framework-based nanomedicines for treating MRSA, TB, and Ctr infections. The authors also showed MOFs as metal ion reservoirs and drug delivery systems for antibacterial agents. Overall, the review was well written and should be accepted after considering including the following comments.

  1. It would be helpful to the readers if a general scheme for making MOFs is described in a dedicated figure.

Re: Thanks for your comments, we add one figure to explain the MOF-based antibacterial nanomedicine. Please check the updated Figure 1.

  1. Can authors include a dedicated section on current limitations for MOFs and perspective at the end?

Re: After careful consideration, we have decided not to include a separate section on current limitations for MOFs and perspective. We believe that the information we have provided in the ‘Conclusions and Prospects’ part is sufficient to convey the main points of our research. Nevertheless, we have made some revisions in the last part to clarify any potential confusion. Thank you again for your valuable feedback, and we appreciate your interest in our research!